# Fibroblast Growth Factors for Nonalcoholic Fatty Liver Disease: Opportunities and Challenges

**DOI:** 10.3390/ijms24054583

**Published:** 2023-02-26

**Authors:** Haoyu Tian, Shuairan Zhang, Ying Liu, Yifan Wu, Dianbao Zhang

**Affiliations:** 1Department of Stem Cells and Regenerative Medicine, Key Laboratory of Cell Biology, National Health Commission of China, and Key Laboratory of Medical Cell Biology, Ministry of Education of China, China Medical University, Shenyang 110122, China; 2Department of Gastroenterology, The First Affiliated Hospital of China Medical University, Shenyang 110001, China

**Keywords:** nonalcoholic fatty liver disease, nonalcoholic steatohepatitis, fibroblast growth factors, FGF-based therapeutics

## Abstract

Nonalcoholic fatty liver disease (NAFLD), a chronic condition associated with metabolic dysfunction and obesity, has reached epidemic proportions worldwide. Although early NAFLD can be treated with lifestyle changes, the treatment of advanced liver pathology, such as nonalcoholic steatohepatitis (NASH), remains a challenge. There are currently no FDA-approved drugs for NAFLD. Fibroblast growth factors (FGFs) play essential roles in lipid and carbohydrate metabolism and have recently emerged as promising therapeutic agents for metabolic diseases. Among them, endocrine members (FGF19 and FGF21) and classical members (FGF1 and FGF4) are key regulators of energy metabolism. FGF-based therapies have shown therapeutic benefits in patients with NAFLD, and substantial progress has recently been made in clinical trials. These FGF analogs are effective in alleviating steatosis, liver inflammation, and fibrosis. In this review, we describe the biology of four metabolism-related FGFs (FGF19, FGF21, FGF1, and FGF4) and their basic action mechanisms, and then summarize recent advances in the biopharmaceutical development of FGF-based therapies for patients with NAFLD.

## 1. Introduction

Due to a sedentary lifestyle and excess nutrition, overweight and obesity, as well as a series of metabolic-related diseases, are developing among modern people [1]. Nonalcoholic fatty liver disease (NAFLD) is closely related to metabolic syndrome and has become the most prevalent chronic liver disease in recent decades [2]. The disease burden of NAFLD varies by geographic region and ethnicity [3]. According to a meta-analysis in 2022, the global prevalence of NAFLD is approximately 30% and increasing [4]. NAFLD refers to a group of diseases, ranging from mild steatosis to severe nonalcoholic steatohepatitis (NASH), that dramatically increase overall mortality. Obesity is a significant risk factor for the progression of NAFLD, in which excess lipids are stored as triglycerides in hepatocytes, resulting in steatosis [5,6]. There is also as a known genetic predisposition to hepatic fat accumulation (e.g., PNPLA3 gene polymorphisms and TM6SF2 gene variations) [7,8]. The overall mortality of NAFLD patients is increased compared to non-NAFLD patients [9]. Cardiovascular disease (CVD) and cancer are the main causes of death in people with NAFLD [10,11]. NASH is a kind of aggressive fatty liver disease distinguished by the presence of hepatocyte ballooning and lobular inflammation with or without fibrosis [12,13]. The severity of fibrosis is a crucial indicator of the long-term prognosis in NASH patients for it progresses in about 40% of NASH patients [14,15,16]. The progressive form of NAFLD and NASH is a leading indication for a liver transplant [17]. In 2020, several expert panels advocated for metabolic-associated fatty liver disease (MAFLD) as a better term to reflect the heterogeneity of NAFLD.

The exact mechanism of NAFLD is yet to be elucidated. Excess lipids are the primary damage factors, and the subsequent effects of pathogenic drivers, including insulin resistance, lipotoxicity, and immune system activation, all contribute to the pathogenesis of NAFLD [18]. NAFLD is thought to develop with early steatosis, which is insufficient in triggering inflammation and fibrosis. As the condition advances, the second hit (via oxidative stress and other factors) aggravates the liver damage [19]. Furthermore, it was found that multiple factors, including hepatic inflammation and the synergistic effect of multiple mechanisms, including oxidative stress, lipid peroxidation, the endotoxin-induced activation of hepatic stellate cells (HSCs), and mitochondrial dysfunction, promoted the progression of NASH to fibrosis, and this hypothesis was dubbed “multi-hit” [20,21]. Although the development of NAFLD varies from person to person, it is generally categorized into four stages [22,23]. The first stage is fat accumulation in the liver, which is believed to be innocuous [24]. The second stage, early NASH (F0: no fibrosis and F1: negligible fibrosis), is characterized by fatty infiltration and liver inflammation. The diagnosis of NASH requires the detection of steatosis, ballooning, and lobular inflammation on the liver biopsy. Pathological alterations in NASH include portal inflammation, polymorphonuclear infiltration, Mallory–Denk bodies, microvacuolar steatosis, and giant mitochondria. The third stage entails chronic liver inflammation and damage, which both promote persistent inflammation and hepatic fibrosis. The fourth stage is cirrhosis (F4), a severe stage of NAFLD/NASH. Patients with advanced fibrosis and cirrhosis are at an increased risk for liver-related complications (i.e., liver decompensation and HCC) and liver-related mortality [25]. In the early stages, liver fibrosis can both progress and regress. The reversal of fibrosis is often observed with weight loss in obese patients with NAFLD [26]. However, with the progressive inflammation and fibrosis of the liver parenchyma with the disruption of the hepatic architecture, aberrant regeneration eventually led to the irreversible loss of liver function [27]. Thus, a timely intervention for the diseases is needed to improve quality of life and reduce liver-related mortality [28].

In recent years, despite the fact that many valuable breakthroughs have been made in the pathogenesis and treatment of NAFLD, the condition still remains challenging to manage. Currently, the treatment for NAFLD is mainly based on lifestyle changes (such as increased exercise; weight loss; and a diet low in calories, cholesterol, saturated fat, and fructose). With regard to the amelioration of weight loss due to NAFLD, a 5% change can improve steatosis, a 7–9% change is able to improve most histopathological changes, while a weight loss of 10% or more can improve fibrosis [29]. However, lifestyle adjustments and weight loss are uneasy for the patients, let alone long-term maintenance [30]. With most advanced diseases and severe fibrosis being rarely curable with lifestyle interventions, appropriate pharmacological interventions remain in high demand. Despite tremendous scientific efforts being made to develop therapeutics, no specific drug for NAFLD/NASH was approved by the Food and Drug Administration (FDA). There are very few available drugs to treat NAFLD, most of which have been withdrawn from the market due to their side effects [31]. Currently, only vitamin E and the proliferator-activated receptor gamma (PPAR-y) ligand pioglitazone are recommended for selected patients by the European and American Association for the Study of the Liver [32]. Several drugs have been investigated in clinical trials for the histopathology of NASH or aggravation of fibrosis, while the translation to clinical applications still requires additional investigations [33,34]. In addition, individual variation among patients makes it difficult for a single treatment to achieve the desired effect in NAFLD/NASH. In consideration of the above, combination therapies and individualized treatment models of drug intervention appear to be better options.

## 2. The Rationale for FGFs in the Treatment of NAFLD

Fibroblast growth factors (FGFs) are a family of 22 signaling proteins that regulate reproduction, development, repairment, and metabolism [34,35,36]. The human FGF family can be classified into seven subfamilies based on sequence homology and phylogenetic divergence. The majority of FGFs are members of the classical FGF subfamilies (FGF1-10, FGF16-18, FGF20, and FGF22), which act as autocrine and paracrine factors that bind to and activate FGF receptors. Moreover, three endocrine members in the FGF subfamily (FGF19, FGF21, and FGF23) are released from the extracellular matrix into the blood and are involved in the metabolism of lipids, carbohydrates, bile acids, as well as phosphates in distant organs [37]. Endocrine FGFs activate their cognate fibroblast growth factor receptors (FGFRs), which operate as coreceptors with the corresponding glycoproteins. Specifically, FGF19 and FGF21 act as crucial bile acid and glycolipid metabolism regulators by binding to the FGFRs/β-Klotho complex. Meanwhile, FGF23 plays a critical role in regulating phosphate and vitamin D in homeostasis by stimulating the FGFR/α-Klotho complex in the kidney and parathyroid gland. Several other FGF members have also been found in preclinical studies to maintain energy homeostasis and regulate glucose/lipid metabolism [38,39]. FGF analogs have been shown to effectively alleviate the pathological states of hepatic steatosis, steatohepatitis, and hepatic fibrosis (Figure 1).

Considering the potent beneficial roles of endocrine FGFs in NAFLD and metabolic diseases, more and more researchers are paying extra attention to the usage of FGFs in various promising therapeutic approaches. The characteristics of several FGFs with potential therapeutic effects in NAFLD are summarized in Table 1. FGFs and their analogs have pleiotropic metabolic effects. A growing number of FGF analogs and/or receptor agonists (particularly for FGF19 and FGF21) are being explored as potential agents for NAFLD and other metabolic-related diseases [40,41]. To date, a variety of FGF-based drugs have shown therapeutic potential in preclinical studies and clinical trials for NAFLD/NASH [42,43,44,45]. FGF19 and FGF21 may be the most promising potent drugs for NAFLD, especially in the relief of liver steatosis, steatohepatitis, and liver fibrosis [46].

### 2.1. FGF19

FGF19 was the first identified member of the endocrine FGF subfamily. FGF19 is induced in the small intestine and colon via the nuclear receptor farnesoid X receptor (FXR) to regulate postprandial bile acid uptake. Other nuclear receptors, such as the vitamin D receptor (VDR), the retinoid X receptor (RXR), and the pregnane X receptor (PXR), can also promote the production of FGF19. Recent studies have suggested that cholesterol and fat-soluble vitamins modulated FGF19 synthesis [47,48]. FGF19 in the circulatory system can act on various metabolic organs, and the most prominent are the liver, adipose tissue, and brain. FGF19 performs its physiological functions by binding to FGFR4 and β-Klotho to form receptor complexes, which are highly enriched in the liver. FGF19 can also prevent bile acid (BA) biosynthesis in hepatocytes by inhibiting the orphan nuclear receptor small heterodimeric partner (SHP)-dependent cholesterol 7α hydroxylase (CYP7A1) [49]. This mechanism was later found to help prevent BA-induced liver injury in NASH. In addition to the above affects, FGF19 is also involved in lipid and carbohydrate metabolism. FGF15 (FGF19 homologue in mice) knockout mice lost their ability to maintain serum glucose homeostasis and exhibited decreased hepatic glycogen content and glucose intolerance, suggesting that FGF19 may stimulate protein and glycogen production in hepatocytes in an insulin-like manner. FGF19 inhibits gluconeogenesis by suppressing the cAMP response element-binding protein (CREB)/peroxisome proliferator-activated receptor γ coactivator 1-α (PGC1α) signaling cascade [50]. However, FGF19 differs from insulin in its biological properties and mechanisms of action. Firstly, FGF19 simultaneously inhibited fatty acid synthesis. Secondly, the function of FGF19 in regulating hepatic protein and glycogen metabolism is not mediated by the activity of the protein kinase Akt but is instead mediated by the mitogen-activated protein kinase signaling pathway [51]. Furthermore, Wu et al. found that FGF19 had a dual involvement in lipid metabolism homeostasis [52], as FGF19 could exert lipid-lowering effects by increasing the elevating energy expenditure and metabolic rate in adipose and other non-hepatic tissues by activating FGFR1c while also increasing lipid levels by activating FGFR4 and downregulating BA synthesis in the liver.

FGF19 plays an essential role in carbohydrate and lipid metabolism and has a favorable effect on NAFLD [53]. FGF19 is a nutrient-regulated postprandial hormone that inhibits gluconeogenesis, while stimulating glycogenesis and protein synthesis without stimulating lipogenesis [54,55]. The activation of the FGF19 signaling pathway is abnormal in NAFLD patients, and fasting and postprandial FGF19 levels are lower [56,57]. Intriguingly, other reports have indicated no significant difference in basal FGF19 levels between NAFLD patients and healthy subjects, whereas patients with liver damage had lower FGF19 levels [58]. In addition, a decrease in FGF19 was detected in NAFLD patients as the condition progresses, along with increases in alanine aminotransferase (ALT), triglyceride (TG), and other indicators [59]. The powerful pharmacological effects of FGF19 on obesity, diabetes, and fatty liver promote the development of FGF19 in clinical applications. Although endogenous and exogenous FGF19 seem prospective for treating NAFLD/NASH, they have some limitations. The upregulation of FGF19 expression is closely associated with hepatocyte dysplasia, tumor formation, and a poor prognosis in individuals with hepatocellular carcinoma (HCC) [60]. The STAT-3 signaling pathway was activated in the hepatocytes of FGF19 transgenic mice via IL-6, transforming normal hepatocytes into malignant cells [61]. However, another investigation showed that FGF15 exposure did not increase the risk of HCC development in mouse models of metabolic dysregulation (db/db, diet-induced obesity, and multidrug resistance-related protein 2 (Mdr2) null mice) [62]. There are specific differences in amino acid sequences between FGF19 and FGF15, which may lead to different signals activated. The increasing risk of HCC by FGF19 results in clinical application difficulties. Therefore, researchers are conducting an in-depth analysis of the structural basis of FGF19 metabolism and mitotic activity, with the expectation of developing FGF19 variants without mitotic potential [63].

### 2.2. FGF21

In 2005, Kharitonenkov et al. found that FGF21 could induce adipocytes to absorb glucose and lower blood glucose. FGF21 generally acts directly or indirectly on multiple major organs, most notably the adipose tissue, liver, and brain. It could protect people from obesity, insulin resistance, aberrant metabolisms, and irregular vascular homeostasis to a certain extent [64]. Liver and adipose tissue are the primary sources of FGF21, which were also detected in the pancreatic islets, skeletal muscle, heart, kidney, and specific other metabolic organs [65]. FGF21 usually recruits β-Klotho and FGFRs (FGFR1c, FGFR2c, or FGFR3c) as coreceptors for activation [66]. FGFRs are widely expressed in humans, whereas β-klotho is only expressed in specific tissues [67]. FGF21 was found to play a significant role in regulating energy homeostasis in preclinical and clinical studies [68]. Physiological doses of FGF21 can reduce body weight and fat content, and also alleviate insulin resistance, hyperglycemia, and dyslipidemia. FGF21 is essential for fasting liver metabolism to drive blood glucose elevation through various pathways, such as fatty acid oxidation [69]. In NAFLD mice, recombinant FGF21 reduced hepatic steatosis by increasing fatty acid oxidation and reducing lipogenesis [70]. Furthermore, FGF21 analogs reduced liver inflammation and fibrosis in NASH mice [71]. FGF21-deficient mice developed steatosis, worsening the inflammation and fibrosis when administered a methionine- and choline-deficient diet [72]. In the liver fibrosis and NASH models in vitro and in vivo, FGF21 receptor agonists inhibited liver inflammation, fat content, and liver fibrosis. Notably, FGF21 has been shown to directly reduce lipid deposition in hepatocytes in a non-insulin-based way, thereby hindering the development of NAFLD. Unlike FGFR1c/β-klotho receptor agonists such as BFKB8488A and MK3655, the mechanism of action of FGF21 polypeptide analogs is more complex than simply activating the FGFR1c/β-klotho receptor complex [73]. The administration of FGF21 has beneficial effects on a range of NAFLD-related complications, including a reduction in fat mass and the alleviation of hyperglycemia, insulin resistance, and NAFLD [74]. Since FGF21 is a metabolically active hormone derived mainly from liver tissue, FGF21-based therapy strategies have drawn significant attention.

Unlike FGF19, FGF21 does not induce the adverse effects of mitogenesis in vivo [40]. In addition, the transgenic expression of FGF21 in mice renders them resistant to obesity and chemically induced hepatocarcinogenesis [64]. However, human-natural FGF21 (hFGF21) cannot be employed directly for pharmacological therapy due to its short half-life (30–90 min), poor biological properties (such as rapid aggregation in soluble formulations), and proteolytic instability. In addition, endogenous FGF21 inactivation enzymes represent major obstacles to the clinical implementation of FGF21-based pharmacotherapies for NAFLD [40]. In this case, various genetic engineering approaches have been applied to develop hFGF21 analogs [75].

### 2.3. FGF1

Fibroblast growth factor 1 (FGF1) has been described in the stellate cells of the regenerating liver [76], which is a mitogenic factor involved in embryonic development, wound healing, neurogenesis, and angiogenesis. In FGF1 knockout mice, hyperglycemia and insulin resistance developed under high-fat diet (HFD) conditions. FGF1 can be regulated by PPARγ in adipose tissue, and induced by feeding and HFD stimulation. The PPARγ-FGF1 axis is necessary for fat homeostasis and systemic insulin sensitivity. In both gene- and diet-induced obese mice, a single dosage of recombinant FGF1 (rFGF1) normalized blood glucose levels without causing hypoglycemia. Furthermore, the long-term administration of rFGF1 promoted skeletal muscle glucose uptake while inhibiting hepatic glucose production, resulting in systemic insulin sensitization and sustained blood glucose reduction. In the NASH mice model, the administration of rFGF1 did not affect collagen deposition but ameliorated liver inflammation and hepatocyte injury [77]. Furthermore, rFGF1 was able to attenuate diabetes-mediated liver fibrosis, hepatocyte steatosis, apoptosis, and other pathological features in db/db mice [78]. In spite of the multiple benefits of FGF1, the increased potential tumorigenic risk of long-term use limits its therapeutic utility. The elevation of FGF1 was observed in various tumors, including prostate cancer and osteosarcoma [79,80]. The rFGF1^ΔNT^ is a variant of FGF1 that lacks the first 24 residues from the N-terminus. This kind of variant reduced the proliferative properties while maintaining an insulin-sensitizing effect. Long-term studies of rFGF1^ΔNT^ are warranted to determine its safety as a therapeutic.

### 2.4. FGF4

FGF4 is another autocrine/paracrine factor that is expressed at low levels in adults’ duodenum, ileum, and colon for the maintenance of intestinal stem cells [81]. FGF4 is critical for wound repair, organ regeneration, and energy metabolism by activating FGFRs [82]. FGF4 has a more restricted FGFR1c binding specificity, followed by FGFR2c, FGFR3c, and FGFR4. Recent studies have shown that FGF4 is essential in regulating glucose and lipid metabolism, as well as maintaining systemic metabolic homeostasis [39]. The administration of FGF4 to mice ameliorated insulin resistance, while also inhibiting the infiltration of fat-associated macrophages and the development of inflammation. FGFR1c was abundantly expressed in macrophages, and AMPKα activation in macrophages ameliorates adipose tissue inflammation [83]. FGF4 acts directly on macrophages to block inflammatory responses in the liver and adipose tissue, to protect against dietary obesity-related NAFLD and stress-induced liver injury [84,85]. In NAFLD patients and mice models, hepatic FGF4 expression was found to be inversely correlated with the pathological grade of NAFLD. FGF4 knockout exacerbates liver steatosis, inflammation, and damage, which can be alleviated by recombinant FGF4. Yet, the tumorigenic risk of the long-term application of FGF4 remains its primary challenge. Ying et al. designed a non-mitotic rFGF4 analog (rFGF4^ΔNT^) that impairs the dimerization and activation of heparan sulfate-assisted FGFRs [39]. In comparison to wild-type FGF4, rFGF4^ΔNT^ retains its metabolic activity while losing mitotic activity. In mice fed a high-fat diet, it was found that rFGF4^ΔNT^ presented significant protective effects on NASH through an AMPK-dependent signaling pathway [86].

## 3. Biopharmaceutical Strategies of FGF-Based Treatment

Over the past few years, FGF-based therapies for NAFLD/NASH have gathered substantial investment. Various clinical trials have revealed that FGF19 and FGF21 analogs effectively reduce hepatic fat content. A number of randomized controlled trials (RCTs) and cohort studies have tested FGF-19 and FGF-21 analogs in patients with NAFLD/NASH (Table 2). Furthermore, several trials are still ongoing to evaluate the efficacy of these drugs (Table 3).

Aldafermin (formerly known as NGM282 or M70) is a synthetic analog of human FGF19, carrying three amino acid substitutions (A30S, G31S, and H33L) and a five-amino acid deletion compared to wild-type FGF19 [89]. Aldafermin is devoid of pro-tumoral activity, but fully retains metabolism regulatory activities [90]. Aldafermin does not activate the STAT3 pathway which is a crucial signaling pathway for FGF19-mediated hepatocarcinogenesis. Aldafermin has shown potent anti-adipose, anti-inflammatory, and anti-fibrotic activities in multiple animal experiments [91,92].

The first FGF21-based molecule to enter clinical development is a modified FGF21 variant, LY2405319 (LY) [93]. Following former use, both type 1 and type 2 diabetes patients experienced significant weight loss and lipid changes, including substantial decreases in TGs and LDLc, and marked increases in HDLc and adiponectin (a key adipokine with insulin-sensitizing, anti-inflammatory, and anti-fibrotic properties) [94]. LY caused a significant drop in TGs as early as on day three of dosing and continued to drop throughout the study. Although the clinical development of LY compounds was interrupted by the lack of the glycemic effect, this study lays the groundwork for the drug development of other molecules of this class [95].

The second clinical experience with FGF21 analogs comes from a long-acting FGF21 variant reported by Pfizer, PF-05231023 (CVX-343) [96]. PF-05231023 is an antibody-conjugated FGF21 analog that has been found in both mice and humans to reduce body weight [96,97]. Subjects with type 2 diabetes experienced persistent weight loss during PF-05231023 dosing. In a follow-up study of obese hypertriglyceridemia patients, PF-05231023 showed favorable lipid alterations [96,98]. However, PF-05231023 has also been found to cause increases in blood pressure and heart rate, which may contribute to the drug’s demise.

Pegbelfermin (BMS-986036) is a pegylated FGF21 analog with a longer half-life [99,100]. In animal experiments, pegbelfermin improved the histological features of NASH and fibrosis, and increased adiponectin levels [101]. Efruxifermin (EFX, AKR001) is a long-acting human immunoglobulin 1 (IgG1) Fc-fused FGF21 analog with promising pharmacological effects in numerous preclinical animal experiments [102]. Its C-terminal region was modified by two amino acid substitutions (P171G and A180E), increasing the affinity of FGF21 for β-Klotho, to elevate the FGFR-mediated signaling activity, functional potency, and timing of efruxifermin action in vivo. Efruxifermin has a longer duration of action than most human FGF21 analogs, with a half-life of 3–3.5 days.

### 3.1. The Effects of FGF Drugs on NAFLD/NASH Histology

Currently, a biopsy remains the most accurate method for assessing the grade of NAFLD and its inflammatory and fibrotic components [103]. The NAFLD activity score (NAS) quantifies disease activity based on histological manifestations of hepatic steatosis, inflammation, and hepatocyte expansion [91,104]. The NAS system ranks each feature, such as steatosis severity from 0 to 3, the severity of hepatocyte expansion from 0 to 2, and the severity of lobular inflammation from 0 to 3. An improvement in histological severity was accompanied by a reduction in NAS values [104]. However, as fibrosis is considered a marker of disease stage rather than injury grade, it is not included in the NAS score [105]. In the NASH Clinical Research Network (CRN) system, stage 0 = no fibrosis; stage 1 = centrilobular or periportal fibrosis; stage 2 = centrilobular and periportal fibrosis; stage 3 = bridging fibrosis; and stage 4 = cirrhosis [106,107].

Several studies have investigated what roles FGF analogs played in human liver histology. In an open-label study, 12 weeks of NGM282 treatment significantly improved histological outcomes in patients with NASH, as evidenced by reductions in NAS and fibrosis scores [44]. Percentages of 50% (NGM282 1 mg) to 68% (NGM282 3 mg) achieved significant histological improvement (two or more reductions in NAS, no worsening or improved fibrosis, or no worsening NASH). More importantly, steatosis, hepatocyte ballooning, and lobular inflammation were significantly improved from baseline in all participants after NGM282 treatment. Later, Harrison et al. used NGM282 for longer-term treatment to more accurately assess the efficacy and safety of the drug [43]. The results show that 24 weeks of NGM282 treatment showed positive results on histological endpoints. Fibrosis improvement and NASH remission without worsening fibrosis were achieved in 38% and 24% of patients, respectively. ALPINE 2/3 is a randomized, double-blind, placebo-controlled Phase 2b study. The study found that approximately 30% of patients with biopsy-proven NASH achieved improvements in fibrosis with 0.3, 1.0, or 3.0 mg of NGM282 [42]. However, this result did not reach statistical significance compared with the placebo group. This may be due to 24 weeks of treatment being too short to improve fibrosis or NASH. In another two clinical trials, NGM282 for 24 weeks resulted in a more remarkable histological improvement in NASH and a reduction in the NAS score than that of other drugs, such as pan-PPAR agonist lanifibranor and semaglutid [108,109]. In a Phase 2a study of efruxifermin, 80 patients were given 28, 50, or 70 mg of efruxifermin, or a placebo subcutaneously once a week based on the stage of hepatic fat fraction (HFF) and fibrosis [110]. After 16 weeks of treatment, a qualitative histological analysis of paired biopsy samples showed fibrosis regression. In total, 70% of patients in the biopsy evaluable analysis set (BAS) group had a ≥two-point improvement in NAS without worsening fibrosis. In the meantime, 48% of patients showed a resolution of NASH without worsening fibrosis.

### 3.2. The Therapeutic Effect of FGF Drugs Evaluated by MRI-PDFF

Liver biopsy remains the gold standard for assessing the grade of NAFLD and assessing disease activity [103]. However, biopsies are limited by sampling error, procedure-related mortality, and cost [91]. Several noninvasive liver tests (NILTs) are relatively accurate at determining the degree of hepatic steatosis and fibrosis. Magnetic resonance imaging-proton density fat fraction (MRI-PDFF) is a noninvasive and highly reproducible technique that measures the amount of fat in the liver, which is less susceptible to the sampling error than a single liver biopsy [108,109]. The best imaging approach at the moment is MRI-PDDF, which can assess triglyceride concentrations in the liver both qualitatively and quantitatively [111]. MRI-PDDF has the highest accuracy (99%) compared to other NILTS [111].

In the initial randomized, double-blind, and placebo-controlled Phase II trial of NGM282 at 3 and 6 mg in patients with biopsy-confirmed NASH, a reduction in liver fat content was found using MRI-PDFF (3 mg NGM282: −12.9% [SE 8.8%], *p* = 0.002; 6 mg NGM282: −18.9% [SE 6.3%] *p* < 0.001) [89]. Among them, patients with a liver fat content (LFC) greater than 20%, measured by baseline MRI-PDFF, had the most considerable reduction. In another clinical trial, most of the beneficial effects of NGM282 in relation to the liver fat content were found as early as week 6 [44]. In addition, 92% (1 mg) and 100% (3 mg) of patients, respectively, achieved a ≥5% reduction in the absolute liver fat content and a ≥30% reduction in the relative liver fat content after 12 weeks of treatment with NGM282. Meanwhile, 33% (1 mg) and 63% (3 mg) of patients had utterly normalized liver fat (defined as absolute LFC ≤5%). In the 24-week study, the study used MRI-PDFF to measure an absolute LFC reduction of 7.7% compared to baseline in patients in the 1 mg NGM282 group, and it was much better than in patients in the placebo group (difference, 5.0%; 95% CI, 8.0% −1.9%; *p* = 0.002). Relative LFC reductions higher than 30% were reported in 66% of the NGM282 group and 29% in the placebo group. Subsequent studies found better LFC reduction rates with 3.0 mg of NGM282 (−11.4% [SE 1.0%]) [42].

Significant reductions in the LFC were achieved by taking 10 mg of pegbelfermin daily or once a week as measured by MRI-PDFF [45]. After 16 weeks of treatment, pegbelfermin decreased the mean LFC by 6.8% (10 mg, *p* = 0.0004) and 5.2% (20 mg, *p* = 0.008), respectively, compared with the placebo. This MRI-PDFF change was similar between patients with type 2 diabetes and those without type 2 diabetes. In addition, treatment with efruxifermin for 12 weeks reduced absolute HFF in NASH patients by 12–14% [88]. At the highest dose of efruxifermin (70 mg), HFF returned to normal in approximately 67% of patients.

### 3.3. Safety Investigation of FGF Drugs

Preclinical studies have shown that alderfermin is well-tolerated by animals given therapeutic doses for up to 26 weeks. Theoretically, extrahepatic adverse reactions (cardiovascular atherosclerotic disease) may occur in subjects with NASH in prolonged administration. Nevertheless, these symptoms were not detected in clinical trials, but the most common adverse reactions of aldafermin were gastrointestinal (diarrhea, abdominal pain, and nausea). The majority of adverse effects during treatment were mild to moderate. For instance, diarrhea is usually short-lived and can be managed by separating drug administration from mealtimes or reducing meal portions [42,44]. There were no sequelae of alderfermin, and the investigator deemed all major adverse events unrelated to the drug.

Pegbelfermin and efruxifermin were also well-tolerated [45,88]. There was no adverse event-related death for either drug. Similar to FGF19 analogs, the main adverse reactions are gastrointestinal reactions, such as diarrhea and nausea. Moreover, one patient experienced worsening depression and attempted suicide after taking 10 mg of pegbelfermin daily. One case of acute pancreatitis complicated by diabetic ketoacidosis (grade 4) in the efruxifermin group was considered to be related to the study drug and led to discontinuation. However, the patient was morbidly obese and severely insulin-resistant prior to the research.

## 4. Concluding Remarks

The increasing prevalence of NAFLD/NASH contributes to the global health burden that has heightened scientific interest in addressing this segment of the population. Although no medications have been licensed by the Federal and Drug Administration (FDA) and the European Medicine Agency (EMA) for the treatment of NAFLD/NASH, several clinical trials have been histologically identified and have confirmed that certain drugs could improve the histology condition. This has encouraged researchers and the pharmaceutical industry to pursue more durable and stable treatment strategies.

Members of the FGF family are essential regulators of energy homeostasis and glucose/lipid metabolism. Increasingly promising mouse and human experiment data support the idea that FGFs (FGF19 and FGF21) can prevent the onset and progression of NAFLD by inhibiting hepatic fat accumulation, excessive inflammation, hepatocyte damage, and fibrosis. Non-mitotic hFGF19 variants and long-acting hFGF21 analogs have been developed for clinical trial evaluation. The analogs and mimetics of FGF19 and FGF21 have been repeatedly observed to improve dyslipidemia and NAFLD in multiple clinical trials. Both of the above can reduce liver fat and improve the degree of liver steatosis and fibrosis. On the other hand, the safety and therapeutic effects of FGF1 and FGF4 in animals and humans need to be further evaluated. At the same time, concerns about tumorigenesis represent another major impediment in developing these two FGFs into metabolized drugs for patients who require lifetime therapy.

The impact of FGF-based therapies is obviously meaningful in order to improve NAFLD/NASH. Despite the fact that the signaling pathways involved in FGF-mediated metabolic improvement are still not being fully explored, significant progress has been made in developing FGF-based therapeutics. The keys to boosting the clinical success rate of novel therapeutic applications are comprehensively carrying out standardized clinical trials and customizing excellent animal models. Simultaneously, given the complexity of NAFLD/NASH pathophysiology, drug intervention with multiple targets and pathways is required to improve the therapeutic effect. More research needs to be conducted in order to better explore the pathogenic mechanisms associated with liver metabolism, inflammatory responses, and injury during the development of FGF and NAFLD/NASH, in turn helping to develop safer and more sustainable treatments.

## Figures and Tables

**Figure 1 ijms-24-04583-f001:**
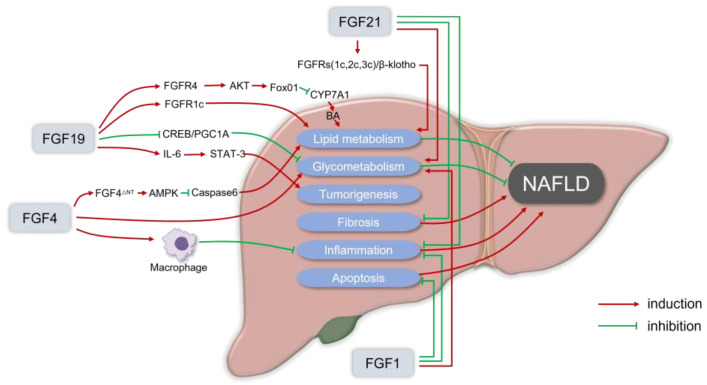
The rationale for fibroblast growth factors (FGFs) in the treatment of nonalcoholic fatty liver disease (NAFLD). FGF19 usually binds to the FGFR1c or FGFR4/β-Klotho complex to enhance lipid metabolism that can alleviate NAFLD. FGF19 inhibits the orphan nuclear receptor small heterodimeric partner (SHP)-dependent cholesterol 7α hydroxylase (CYP7A1) to prevent BA-induced liver injury in NASH. For glycometabolism, FGF19 inhibits gluconeogenesis by suppressing the cAMP response element-binding protein (CREB)/peroxisome proliferator-activated receptor γ coactivator 1-α (PGC1α) signaling cascade. Meanwhile, in FGF19 transgenic mice, it was observed that FGF19 could transform normal hepatocytes into malignant cells via IL-6 by activating the STAT-3 pathway. FGF21 usually recruits β-Klotho and FGFRs (FGFR1c, FGFR2c, or FGFR3c) as coreceptors for activation, to reduce lipid deposition in hepatocytes in a non-insulin way, and the FGF21 receptor agonists can inhibit liver inflammation, fat content, and liver fibrosis in the in vitro and in vivo models of liver fibrosis and NASH. Physiological doses of FGF21 can reduce the body weight, and fat content can also alleviate insulin resistance, hyperglycemia, and dyslipidemia. In the NASH mice model, the administration of recombinant FGF1 (rFGF1) ameliorated liver inflammation and hepatocyte injury, and FGF1 normalized the blood glucose levels of both gene- and diet-induced obese mice. FGF4 is also essential in regulating glucose and lipid metabolism, as well as maintaining systemic metabolic homeostasis, which can ameliorate insulin resistance and inhibit the infiltration and the development of inflammation. FGF4 acts directly on macrophages to block inflammatory responses in the liver and adipose tissue. rFGF4^ΔNT^, a non-mitotic rFGF4 analog, can have a significant protective effect on NASH through an AMPK-dependent signaling pathway.

**Table 1 ijms-24-04583-t001:** Major physiological/pathological properties of FGF-19, FGF-21, FGF-1, and FGF-4.

FGF Family	FGFs	Receptor	Major Function
Endocrine FGFs	FGF-19	FGFR1c,2c,3c,4	Lipid, bile acid, and energy metabolism
	FGF-21	FGFR1c,3c	Lipid and energy metabolism; insulin sensitivity; and glucose homeostasis
Canonical FGFs	FGF-1	All FGFRs	Adipose tissue homeostasis
	FGF-4	FGFR1c,2c,3c,4	Wound repair, angiogenesis, and energy metabolism

FGFs: fibroblast growth factors; FGFR: fibroblast growth factor receptor.

**Table 2 ijms-24-04583-t002:** A list of FGF-19 and FGF-21 analogs in clinical trials for the management of NAFLD/NASH.

	Year	Refs	Phase	Conditions	Intervention(s)	Duration (Week)	Outcomes
FGF-19 analog							
Aldafermin (NGM282)	2018	[87]	2	biopsy-confirmed NASH, NAS ≥ 4, stage 1-3 fibrosis, absolute LFC ≥ 8% (*n* = 82)	NGM282 6 mg vs. NGM282 3 mg vs. placebo	12	74% patients in the 3 mg dose group and 79% in the 6 mg dose group achieved at least a 5% reduction in absolute LFC from baseline
Aldafermin (NGM282)	2020	[44]	2	biopsy-confirmed NASH, NAS ≥ 4, stage 1–3 fibrosis, absolute LFC ≥ 8% (*n* = 43)	NGM282 3 mg vs. NGM282 1 mg	12	Histological features of NASH with significant reductions in NAS and fibrosis scores, accompanied by improvements in noninvasive imaging and serum markers
Aldafermin (NGM282)	2021	[43]	2	biopsy-proven NASH, NAS ≥ 4, stage 2 or 3 fibrosis, absolute LFC ≥ 8% (*n* = 78)	aldafermin 1 mg vs. placebo	24	Aldafermin reduced liver fat and produced a trend toward fibrosis improvement
Aldafermin (NGM282)	2022	[42]	2b	biopsy-confirmed NASH, stage 2 or 3 fibrosis (*n* = 171)	aldafermin 0.3 mg vs. aldafermin 1 mg vs. aldafermin 3 mg vs. placebo	24	Aldafermin induced a robust imaging and biochemical response in patients with NASH and F2 or F3 fibrosis; no significant dose response of aldafermin on the histological fibrosis primary endpoint
FGF-21 analog							
Pegbelfermin	2018	[49]	2a	biopsy-confirmed NASH, BMI ≥ 25 kg/m2, fibrosis stage 1-3, HFF ≥ 10% (*n* = 184)	pegbelfermin (10 mg or 20 mg) vs. placebo	16	Pegbelfermin significantly reduced HFF in patients with NASH
Efruxifermin	2021	[88]	2a	biopsy-confirmed NASH, fibrosis stage 1–3, NAS of ≥4, HFF of ≥10% (*n* = 80)	efruxifermin (28 mg, 50 mg, or 70 mg) vs. placebo	16	Efruxifermin significantly reduced HFF in patients with F1–F3 stage NASH

BMI: body mass index; FGF-19: fibroblast growth factor-19; FGF-21: fibroblast growth factor-21; HFF: hepatic fat fraction; LFC: liver fat content; NAS: NAFLD activity score; NASH: nonalcoholic steatohepatitis.

**Table 3 ijms-24-04583-t003:** Ongoing clinical human studies concerning the potential efficacy of FGF19 and FGF21 on NAFLD/NASH.

Medication	Characteristics	Trial Num.	Phase	Patients	Period (Week)	Primary Outcomes
NGM282 (Aldafermin)	FGF19 analog	NCT04210245	2b	Compensated Cirrhosis, NASH	48	Improvement in ELF score; safety assessed by reported and observed adverse events
efruxifermin	FGF21 analog	NCT05039450	2b	NASH	36	Change from baseline in fibrosis with no worsening steatohepatitis assessed by the NASH CRN system
efruxifermin	FGF21 analog	NCT04767529	2b	NASH	24	Change from baseline in liver fibrosis with no worsening steatohepatitis assessed by the NASH CRN system
BIO89-100	FGF21 analog	NCT04048135	1, 2	NASH	20	To characterize the effect of BIO89-100 on liver histology by improving the NAS score
BOS-580	FGF21 analog	NCT04880031	2a	NASH	12	The effects of BOS-580 on safety and tolerability will be assessed
BIO89-100	FGF21 analog	NCT04929483	2	NASH	24	The proportion of participants with histological resolution or ≥1 stage decreases in fibrosis stage of NASH without worsening of fibrosis

CRN: clinical research network; ELF: enhanced liver fibrosis; FGF-19: fibroblast growth factor-19; FGF-21: fibroblast growth factor-21; NAS: NAFLD activity score; NASH: nonalcoholic steatohepatitis.

## Data Availability

Data sharing not applicable.

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
