# Peer review of "Fibroblast Growth Factors for Nonalcoholic Fatty Liver Disease: Opportunities and Challenges"

_ijms, 2023, doi:10.3390/ijms24054583_

Round 1

Reviewer 1 Report

The review article by Tian and coworkers addresses a timely and important topic that should be of considerable interest to the scientific community and journal’s readership.  It cites the relevant literature extensively, albeit somewhat selectively at times.  The focus on the FGF isoforms known to play a role in metabolic homeostasis is justified and well-reasoned.  There are, however, a handful of issues with the document (some significant, some minor) that need to be addressed to enhance the manuscript.

The review does a solid job of citing relevant literature, but frequently lacks an interpretation by the authors to provide context for the reader.  This is important because as a review, it should seek to inform a readership not necessarily well acquainted with the subject matter.

There are numerous occasions where statements in the narrative are misleading or confusing.  For instance, line 36 states that NAFLD affects >25% of the world population whilst citing a paper focused on the US demographic.  In lines 42-45, the manuscript implies that CVD is the primary cause of NAFLD-mediated deaths, and that liver transplantation (in addition to cancer) is responsible for end-stage liver disease-related deaths.  Neither sentiment is accurate.  These sentences need to be reworded to accurately convey the intended meaning.  This issue extends to a general issue with the written English throughout—specifically, the poor implementation of punctuation, syntax, and tense.  Since English is the lingua franca here, the authors are strongly encouraged to seek critical proof-reading of the document of any future revision.

In section 2.1 (FGF19), the authors should state that FGF15 (murine) is the ortholog to the human FGF19.  Not doing so, renders the discussion confusing to the uninitiated.

The authors laud the potential therapeutic benefits of the FGF formulations (notably FGF21), but provide only a modest acknowledgement of the clinical issues and limitations to date.  These are well described in Geng et al. (2020, Nat. Rev. Endocrinology, 16:654-667).  The review would be strengthened by a more balanced synopsis.

In line 69, the authors are encouraged to distinguish between liver decompensation and cancer (HCC) when discussing end-stage liver disease.

The manuscript would be enriched by a clearer account of how NAFLD (which is reversible) can progress to end-state fibrosis (cirrhosis) which is irreversible and what that means for the therapeutic interventions being discussed.

In several places the authors refer to “agonists of FGF”.  This is inaccurate (and confusing) because FGF is the agonist for FGFRs, as are the FGF analogs being tested as therapeutics.

In line 224, the authors reference drugs without providing a description of their targets, hence, the significance of the statements made is lost unless the reader aware of their function.

In lines 228-230, the authors note that FGF21 signaling is positively associated with NAFLD.  This flies in the face of the rationale for using FGF21 therapeutically, and is confusing unless the reader is made aware of the idea of FGF21 resistance.  This issue should be discussed in order to provide context for the statement.

Line 234 in contradictory.  How can FGF21 be both non-mitotic and oncogenic in vivo?  Likewise, lines 242-247 are confusing, and contains a seemingly incomplete thought.  The review article ostensibly focuses on FGF functions in NAFLD, so statements about FGF1 activities in other developmental events in extraneous and superfluous.  The sentence beginning with “Contrary to the…” seems to harbor an incomplete thought.  This speaks to the previous comment about sentence structure and rigorous proofing of the document.

Line 259, “lacks the first 24 amino-terminal residues” is tautological.  By definition, the N-terminal residues are the first.  Line 266, HGF isn’t “hepatocyte growth-promoting factors”.  Also there’s no need to abbreviate anything if it’s only stated once in the document.

Table 1 is confusing to interpret.  Poor tab-delineation makes it difficult to decipher which columns pertain to which FGFs and the relevant drugs.  The values presented under “changes versus baseline” are unclear in their meaning.  This also applies to Figure 2 for the liver fat contents.  As presented, the reduced values are not accurately presented as a percent reduction of the baseline levels, nor as an absolute percentage of the new liver fat content.  As presented, the changes are not actual percentages.

Line 294, what is the precise change in the N-terminus of the FGF19 analog?

The paragraphs (lines 301-326) describe the metabolic observations reported on using FGF21-related compounds, but its unclear how this narrative related to NAFLD.  Most of the weight loss effects are due to FGF21 signaling in white adipose deposits.

Line 368, the authors criticize liver biopsies in lieu of imaging modalities for staging liver disease as inadequate due to the small sample size (ie. incomplete surveillance, which is correct), but make no mention of the clinical risks due to its invasiveness.

Line 388, in stating a return to “utterly normalized liver fat”, what is that percentage?

Line 409, the reference to “by separated from” presumably means drug administration.  This should be clarified.

Line 430, the authors should stipulate which FGFs are being referred to.

Lines 440-441, the use of ‘meaningful” twice in the same sentence is redundant.

Author Response

Dear Editor and Reviewers,

Thank you very much for your kind processing and professional comments. The manuscript has been improved following the helpful suggestions. The changes were marked using built-in Track Changes in Microsoft Word and the main modifications are as following:

Reviewer 1:

The review article by Tian and coworkers addresses a timely and important topic that should be of considerable interest to the scientific community and journal’s readership.  It cites the relevant literature extensively, albeit somewhat selectively at times.  The focus on the FGF isoforms known to play a role in metabolic homeostasis is justified and well-reasoned.  There are, however, a handful of issues with the document (some significant, some minor) that need to be addressed to enhance the manuscript.

Response:

Thank you for your careful checking and professional comments. Multiple studies have found the regulation of metabolic homeostasis by FGFs, and our previous studies have also found that some peptides have similar effects. Their promising application prospects motivate us to investigate the challenges we face. Thus, we review the research progress of FGF isoforms in NAFLD, with a view to provide a reference for subsequent research. However, we are still rudimentary in this field, and it is also hoped that our later studies will provide more valuable data that deserve the attention of you and experts in the field. Thanks for your helpful comments, we learned and revised accordingly.

The review does a solid job of citing relevant literature, but frequently lacks an interpretation by the authors to provide context for the reader.  This is important because as a review, it should seek to inform a readership not necessarily well acquainted with the subject matter.

Response:

It is really important to inform a cross-subject readership. Some context has been added in the revised manuscript following your suggestion.

There are numerous occasions where statements in the narrative are misleading or confusing.  For instance, line 36 states that NAFLD affects >25% of the world population whilst citing a paper focused on the US demographic.  In lines 42-45, the manuscript implies that CVD is the primary cause of NAFLD-mediated deaths, and that liver transplantation (in addition to cancer) is responsible for end-stage liver disease-related deaths.  Neither sentiment is accurate.  These sentences need to be reworded to accurately convey the intended meaning.  This issue extends to a general issue with the written English throughout—specifically, the poor implementation of punctuation, syntax, and tense.  Since English is the lingua franca here, the authors are strongly encouraged to seek critical proof-reading of the document of any future revision.

Response:

Thank you for your careful checking. Following your suggestions, line 36 has been corrected to “The disease burden of NAFLD varies by geographic region and ethnicity [3]. According to a meta-analysis in 2022, the global prevalence of NAFLD is approximately 30% and in-creasing [4].” And the referenced has been changed to:

  1. Younossi, Z.M.; Golabi, P.; Paik, J.M.; Henry, A.; Van Dongen, C.; Henry, L. The global epidemiology of nonalcoholic fatty liver disease (NAFLD) and nonalcoholic steatohepatitis (NASH): a systematic review. Hepatology 2023, doi:10.1097/hep.0000000000000004.
  2. Yip, T.C.; Vilar-Gomez, E.; Petta, S.; Yilmaz, Y.; Wong, G.L.; Adams, L.A.; de Lédinghen, V.; Sookoian, S.; Wong, V.W. Geo-graphical similarity and differences in the burden and genetic predisposition of NAFLD. Hepatology 2022, doi:10.1002/hep.32774.

In line 42-45, it has been modified to “Once NAFLD is diagnosed, overall mortality is increased compared to non-NAFLD patients [9]. Cardiovascular disease (CVD) is the leading cause of death in people with NAFLD [10]. Furthermore, cancer-related mortality is among the leading causes of mortality in NAFLD patients [11].”

Following your suggestion, the manuscript has been checked and revised by native English editor, to improve the written English.

In section 2.1 (FGF19), the authors should state that FGF15 (murine) is the ortholog to the human FGF19.  Not doing so, renders the discussion confusing to the uninitiated.

Response:

Thank you for your checking, it has been improved to “FGF15 (FGF19 homologue in mice) knockout mice lost their ability to maintain serum glucose homeostasis and exhibited decreased hepatic glycogen content and glucose intolerance”.

The authors laud the potential therapeutic benefits of the FGF formulations (notably FGF21), but provide only a modest acknowledgement of the clinical issues and limitations to date.  These are well described in Geng et al. (2020, Nat. Rev. Endocrinology, 16:654-667).  The review would be strengthened by a more balanced synopsis.

Response:

Thank you for your suggestion. The bottlenecks and challenges we face should be highlighted. Some discussions have been added, such as “In the meantime, endogenous FGF21 inactivation enzymes represent major obstacles to clinical implementation of FGF21-based pharmacotherapies for NAFLD [40].”

In line 69, the authors are encouraged to distinguish between liver decompensation and cancer (HCC) when discussing end-stage liver disease.

Response:

Accordingly, it has been improved to “The fourth stage is cirrhosis (F4), a severe stage of NAFLD/NASH. Patients with advanced fibrosis and cirrhosis are at an increased risk for liver-related complications (i.e., liver decompensation and HCC) and liver-related mortality [25].”.

The manuscript would be enriched by a clearer account of how NAFLD (which is reversible) can progress to end-state fibrosis (cirrhosis) which is irreversible and what that means for the therapeutic interventions being discussed.

Response:

Thanks for your suggestion, the 2nd paragraph in Introduction has been improved: “The fourth stage is cirrhosis (F4), a severe stage of NAFLD/NASH. Patients with advanced fibrosis and cirrhosis are at an increased risk for liver-related complications (i.e., liver de-compensation and HCC) and liver-related mortality [25]. In the early stages, liver fibrosis can both progress and regress. The reversal of fibrosis is often observed with weight loss in obese patients with NAFLD [26]. However, with the progressive inflammation and fibro-sis of the liver parenchyma with the disruption of hepatic architecture, aberrant regeneration eventually led to the irreversible loss of liver function [27]. Thus, timely intervention for the diseases is needed to improve quality of life and reduce liver-related mortality [28].”

In several places the authors refer to “agonists of FGF”.  This is inaccurate (and confusing) because FGF is the agonist for FGFRs, as are the FGF analogs being tested as therapeutics.

Response:

Thank you for your checking, “agonists of FGF” has been corrected to “agonists of FGF receptor” in the revised version of the manuscript.

In line 224, the authors reference drugs without providing a description of their targets, hence, the significance of the statements made is lost unless the reader aware of their function.

Response:

In the revised manuscript, it has been improved to “Unlike FGFR1c/β-klotho receptor agonists such as BFKB8488A and MK3655, the mechanism of action of FGF21 polypeptide analogs is more complex than simply activating the FGFR1c/β-klotho receptor complex [73].”

In lines 228-230, the authors note that FGF21 signaling is positively associated with NAFLD.  This flies in the face of the rationale for using FGF21 therapeutically, and is confusing unless the reader is made aware of the idea of FGF21 resistance.  This issue should be discussed in order to provide context for the statement.

Response:

After review of the study data, these sentences have been changed to “The administration of FGF21 has beneficial effects on a range of NAFLD-related complications, including a reduction in fat mass and the alleviation of hyperglycemia, insulin resistance, and NAFLD [74].”

Line 234 in contradictory.  How can FGF21 be both non-mitotic and oncogenic in vivo?  Likewise, lines 242-247 are confusing, and contains a seemingly incomplete thought.  The review article ostensibly focuses on FGF functions in NAFLD, so statements about FGF1 activities in other developmental events in extraneous and superfluous.  The sentence beginning with “Contrary to the…” seems to harbor an incomplete thought.  This speaks to the previous comment about sentence structure and rigorous proofing of the document.

Response:

According to your comments, the sentence “Unlike FGF19, FGF21 does not induce mitosis and is oncogenic in vivo. FGF21 transgenic mice are also resistant to chemically induced liver tumors” has been changed to “Unlike FGF19, FGF21 does not induce the adverse effects of mitogenesis in vivo [75]. In addition, the transgenic expression of FGF21 renders mice resistant to obesity and chemically induced hepatocarcinogenesis [76].” The sentence “Contrary to the above-mentioned changes, none of the FGF1-knockout mice were presented” has been removed.

Line 259, “lacks the first 24 amino-terminal residues” is tautological.  By definition, the N-terminal residues are the first.  Line 266, HGF isn’t “hepatocyte growth-promoting factors”.  Also there’s no need to abbreviate anything if it’s only stated once in the document.

Response:

As suggested, “lacks the first 24 amino-terminal residues” has been corrected to “lacks the first 24 residues from the N-terminus”. The sentence “Exogenous FGF4 is relevant to liver regeneration, and hepatocyte growth-promoting fac-tors (HGF) treatment promotes the transdifferentiation of mesenchymal stem cells (MSCs) into hepatocyte-like cells” has been improved to “FGF4 is critical for wound repair, organ regeneration, and energy metabolism by activating FGFRs [84]”.

Table 1 is confusing to interpret.  Poor tab-delineation makes it difficult to decipher which columns pertain to which FGFs and the relevant drugs.  The values presented under “changes versus baseline” are unclear in their meaning.  This also applies to Figure 2 for the liver fat contents.  As presented, the reduced values are not accurately presented as a percent reduction of the baseline levels, nor as an absolute percentage of the new liver fat content.  As presented, the changes are not actual percentages.

Response:

The two tables have revised following your suggestions, and the main text in section 3 has been improved accordingly: “A number of randomized controlled trials (RCTs) and cohort studies have tested FGF-19 and FGF-21 analogues in patients with NAFLD/NASH (Table 2). Furthermore, several tri-als are still ongoing to evaluate the efficacy of these drugs (Table 3).”

Line 294, what is the precise change in the N-terminus of the FGF19 analog?

Response:

It has been improved to “Aldafermin (formerly known as NGM282 or M70) is a synthetic analogue of human FGF19, carrying 3 amino acid substitutions (A30S, G31S, and H33L) and a 5-amino acid deletion compared to wild-type FGF19 [90]. Aldafermin is devoid of protumoral activity, but fully retains regulatory activity [91].”

The paragraphs (lines 301-326) describe the metabolic observations reported on using FGF21-related compounds, but its unclear how this narrative related to NAFLD.  Most of the weight loss effects are due to FGF21 signaling in white adipose deposits.

Response:

Thanks for your comments. The purpose of this paragraph is to describe the effects of FGF analogues on NAFLD and related disorders, thus, we have revised the title from “Clinical Trials of FGF-Based Treatment of NAFLD” to “Biopharmaceutical strategies of FGF-Based Treatment.”

Line 368, the authors criticize liver biopsies in lieu of imaging modalities for staging liver disease as inadequate due to the small sample size (ie. incomplete surveillance, which is correct), but make no mention of the clinical risks due to its invasiveness.

Response:

Accordingly, it has been improved to “Liver biopsy remains the gold standard for assessing the grade of NAFLD and assessing disease activity [104]. However, biopsy is limited by sampling error, procedure-related mortality, and cost [105].”

Line 388, in stating a return to “utterly normalized liver fat”, what is that percentage?

Response:

Following your suggestion, “defined as absolute LFC ≤5%” has been added for “utterly normalized liver fat”.

Line 409, the reference to “by separated from” presumably means drug administration.  This should be clarified.

Response:

It has been changed to “For instance, diarrhea is usually short-lived and can be managed by separating drug administration from mealtimes or reducing meal portions [42,44].”

Line 430, the authors should stipulate which FGFs are being referred to.

Response:

“FGF19 and FGF21” has been added to the sentence.

Lines 440-441, the use of ‘meaningful” twice in the same sentence is redundant.

Response:

Thank you for your checking. The sentence has been corrected to “The impact of FGF-based therapy is obviously meaningful in order to improve NAFLD/NASH”.

Reviewer 2 Report

This review summarized the information on targeting FGFs in the treatment of NAFLD/NASH; however, there are several points to be revised.

- Compared to a recent review Frontiers | FGF19 and FGF21 for the Treatment of NASH—Two Sides of the Same Coin? Differential and Overlapping Effects of FGF19 and FGF21 From Mice to Human (frontiersin.org), or other similar articles, what are the stronger points or clear difference when compared with author's manuscript.

- The abstract can be revised. Currently, a half of the abstract introduces on NAFLD. It would be better to contribute a major part to highlight information of FGF derivatives (role and types of FGFs, promising candidates, etc.).

- It would be better to clarify ambiguous statements in the whole manuscript. For an example, "Only vitamin E and pioglitazone are approved for specific patient groups", so what the specific patient groups are? This information would help readers to immediately understand the statement.

- What are the cells responsible for producing FGFs? What are the signaling pathway following FGFR activations? This could be systematically introduced before going to in-depth detail.

- A table to summarize classes and physiological/pathological properties of FGFs may be needed.

- I cannot see the first abbreviation of FGFR. How about its subtypes and expressions?

- Some typos can be seen, such as "Specifically, FGF19 and FGF21 act as crucial bile acid and glycolipid metabolism regulators by binding to the FGFRs/β-Klotho complex While FGF23 plays a critical role in regulating phosphate and vitamin D in homeostasis by stimulating the FGFR/α-Klotho complex in the kidney and parathyroid gland." This looks like 2 sentences. Similar errors can be seen throughout the manuscript.

- Figure 1 could be extensively improved. What are the meanings of colors (red and green) and head (inhibition and induction) of all arrows. These infos should be clear. I would recommend simplifying the figure. The 3D structures of FGFs are not necessary. We know that FGF could activate FGFR, but what FGFs act on CYP7A1, STAT, AMPK, since they are not a receptor? Then how CYP7A1 affect BA? How can current therapies improve NAFLD? In fact, they would involve in the metabolism, fibrosis, and so on, in the induction phase of NAFLD. Now, this picture implies like they have a special pathway. FGF1 has a direct effect on fibrosis? These are just an example to help the authors to improve. Other similar points should be concerned. If authors could not simplify, separate into several figs (each represent individual FGF) could be an option.

 - What is the association between FGF4 and HGF?

- FGF4 acts directly on macrophage, but also via FGFRs?

- All abbreviations should be indicated under the tables?

- Table 1 = Finished RCTs while Table 2 = Ongoing RCTs? This info must be clear.

- I recommend arranging both tables, especially table 2. Since the drug names are the most important thing, they should be in in the first column (not the row). Other corresponding infos could be adjust accordingly.

- "aldafermin still selectively activates FGFR4 potentially by inhibiting the expression of CYP7α1". What is the meaning of this statement? 

- A lot of errors must be corrected such as line 314-317 ,358-356, baseline in Fig2,

- No SD/SEM in Fig2.

- Overall, although the content is appropriate for the publication, the arrangement and writing are required to be improved. Also, the strong point of this article should be emphasized.

Author Response

Dear Editor and Reviewers,

Thank you very much for your kind processing and professional comments. The manuscript has been improved following the helpful suggestions. The changes were marked using built-in Track Changes in Microsoft Word and the main modifications are as following:

Reviewer 2:

This review summarized the information on targeting FGFs in the treatment of NAFLD/NASH; however, there are several points to be revised.

Response:

Thank you for checking our manuscript and providing professional comments. Several research groups and our own research have found that peptides including FGFs play an important role in the regulation of metabolic homeostasis. However, the clinical application of FGFs still faces many challenges. In this manuscript, we reviewed the research progress of FGFs in NAFLD, hoping to provide a basis and ideas for future research. Thanks to your helpful comments and suggestions, the manuscript has been revised and improved accordingly.

- Compared to a recent review Frontiers | FGF19 and FGF21 for the Treatment of NASH—Two Sides of the Same Coin? Differential and Overlapping Effects of FGF19 and FGF21 From Mice to Human (frontiersin.org), or other similar articles, what are the stronger points or clear difference when compared with author's manuscript.

Response:

In this manuscript, we review multiple isoforms of FGFs, including FGF19, FGF21, and others. Both the rationale of action and the progressions of clinical research have been summarized and compared, with the hope of providing references to relevant studies.

- The abstract can be revised. Currently, a half of the abstract introduces on NAFLD. It would be better to contribute a major part to highlight information of FGF derivatives (role and types of FGFs, promising candidates, etc.).

Response:

Thank you for your helpful suggestion, and the Abstract has been improved:

“Abstract: Nonalcoholic fatty liver disease (NAFLD), a chronic condition associated with metabolic dysfunction and obesity, has reached epidemic proportions worldwide. Although early NAFLD can be treated with a lifestyle change, the treatment of advanced liver pathology, such as nonal-coholic steatohepatitis (NASH), remains a challenge. There are currently no FDA-approved drugs for NAFLD. Fibroblast growth factors (FGFs) play essential roles in lipid and carbohydrate me-tabolism and have been recently emerged as promising therapeutic agents for metabolic diseases. Among them, endocrine members (FGF19 and FGF21) and classical members (FGF1 and FGF4) are key regulators of energy metabolism. FGF-based therapies have shown therapeutic benefits in patients with NAFLD, and substantial progress has recently been made in clinical trials. These FGF analogs are effective in alleviating steatosis, liver inflammation, and fibrosis. In this review, we will describe the biology of four metabolism-related FGFs (FGF19, FGF21, FGF1, and FGF4) and their basic action mechanisms, and then summarize recent advances in the biopharmaceutical development of FGF-based therapies for patients with NAFLD.”

- It would be better to clarify ambiguous statements in the whole manuscript. For an example, "Only vitamin E and pioglitazone are approved for specific patient groups", so what the specific patient groups are? This information would help readers to immediately understand the statement.

Response:

Following your suggestion, the manuscript has been checked and revised. For an example, “Only vitamin E and pioglitazone are approved for specific patient groups” has been improved to “Currently, only vitamin E and the proliferator-activated receptor gamma (PPAR-y) ligand pioglitazone are recommended for selected patients by the European- and American Association for the Study of the Liver [32]”.

- What are the cells responsible for producing FGFs? What are the signaling pathway following FGFR activations? This could be systematically introduced before going to in-depth detail.

Response:

Thank you for your suggestions. These are important basic information, and the production of different isoforms of FGFs, their target organs, and their modes of action are introduced in separate sections.

- A table to summarize classes and physiological/pathological properties of FGFs may be needed.

Response:

Following your suggestion, a table (Table 1 in the revised version) has been added to the manuscript to summarize classes and physiological/pathological properties of FGFs.

- I cannot see the first abbreviation of FGFR. How about its subtypes and expressions?

Response:

As suggested, “fibroblast growth factor receptors (FGFRs)” has been added in section 2. FGFRs are a family of receptor tyrosine kinases broadly expressed on the cell membrane. The human FGFR family consists of four members: FGFR1 to FGFR4, sharing high homology.

- Some typos can be seen, such as "Specifically, FGF19 and FGF21 act as crucial bile acid and glycolipid metabolism regulators by binding to the FGFRs/β-Klotho complex While FGF23 plays a critical role in regulating phosphate and vitamin D in homeostasis by stimulating the FGFR/α-Klotho complex in the kidney and parathyroid gland." This looks like 2 sentences. Similar errors can be seen throughout the manuscript.

Response:

Thanks to your checking and comments. The manuscript has been revised accordingly. Besides, the manuscript has been revised by native English editors to improve the English style.

- Figure 1 could be extensively improved. What are the meanings of colors (red and green) and head (inhibition and induction) of all arrows. These infos should be clear. I would recommend simplifying the figure. The 3D structures of FGFs are not necessary. We know that FGF could activate FGFR, but what FGFs act on CYP7A1, STAT, AMPK, since they are not a receptor? Then how CYP7A1 affect BA? How can current therapies improve NAFLD? In fact, they would involve in the metabolism, fibrosis, and so on, in the induction phase of NAFLD. Now, this picture implies like they have a special pathway. FGF1 has a direct effect on fibrosis? These are just an example to help the authors to improve. Other similar points should be concerned. If authors could not simplify, separate into several figs (each represent individual FGF) could be an option.

Response:

Thank you for your comments, and Figure 1 has been revised accordingly. The meaning of the arrow is annotated in the figure. The 3D structures of FGFs have been removed. With respect to the mechanism of action, it is indeed challenging to describe in detail in one figure. The mechanisms involved are well summarized in various review articles, and the main focus here is on the important nodes and the final implications.

- What is the association between FGF4 and HGF?

Response:

Thanks for your checking. “Exogenous FGF4 is relevant to liver regeneration, and hepatocyte growth-promoting factors (HGF) treatment promotes the transdifferentiation of mesenchymal stem cells (MSCs) into hepatocyte-like cells” has been replaced by “FGF4 is critical for wound repair, organ regeneration, and energy metabolism by activating FGFRs [84].”

- FGF4 acts directly on macrophage, but also via FGFRs?

Response:

“FGF4 has more restricted FGFR1c binding specificity, followed by FGFR2c, FGFR3c, and FGFR4” and “FGFR1c was abundantly expressed in macrophages and AMPKα activation in macro-phages ameliorates adipose tissue inflammation [85]” have been added to section 2.4.

- All abbreviations should be indicated under the tables?

Response:

As suggested, all abbreviations have been added under the tables.

- Table 1 = Finished RCTs while Table 2 = Ongoing RCTs? This info must be clear.

Response:

The statement “A number of randomized controlled trials (RCTs) and cohort studies have tested FGF-19 and FGF-21 analogues in patients with NAFLD/NASH (Table 2). Furthermore, several tri-als are still ongoing to evaluate the efficacy of these drugs (Table 3)” has been added to section 3.

- I recommend arranging both tables, especially table 2. Since the drug names are the most important thing, they should be in in the first column (not the row). Other corresponding infos could be adjust accordingly.

Response:

Thanks to your suggestions, the tables has revised and improved accordingly.

- "aldafermin still selectively activates FGFR4 potentially by inhibiting the expression of CYP7α1". What is the meaning of this statement?

Response:

We have reviewed the research data and removed redundant statements. The sentences “Aldafermin (previously known as NGM282 or M70) is a synthetic analog of human FGF19 consisting of 190 amino acids, that differs from wild-type FGF19 at the amino ter-minus. In spite of the differences, aldafermin still selectively activates FGFR4 potentially by inhibiting the expression of CYP7α1” has been replaced by “Aldafermin (formerly known as NGM282 or M70) is a synthetic analogue of human FGF19, carrying 3 amino acid substitutions (A30S, G31S, and H33L) and a 5-amino acid deletion compared to wild-type FGF19 [90]. Aldafermin is devoid of pro-tumoral activity, but fully retains regulatory activity [91].”

- A lot of errors must be corrected such as line 314-317 ,358-356, baseline in Fig2,

Response:

The manuscript has checked and revised accordingly.

- No SD/SEM in Fig2.

Response:

As suggested, SD has been added to the figure.

- Overall, although the content is appropriate for the publication, the arrangement and writing are required to be improved. Also, the strong point of this article should be emphasized.

Response:

Thank you for your comments, which are very helpful to this manuscript and to us. The manuscript has been revised according to your suggestions, and its accuracy, systematism and readability have been improved.

Round 2

Reviewer 2 Report

Honestly, I don't think that the manuscript is thoroughly improved. Here are some examples. 

- Figure 1: I agree that it's so hard to describe everything in just a figure. That is the reason I suggest separating the figure. Also, it is fine to suggest the reader to find and search other publication to seek the detail. However, it is not fair to provide ambiguous or incorrect information. An example on glycometabolism, figure 1 indicate that FGF4 induces this process, but FGF19 inhibits it; but both of them inhibit NAFLD. Dose this figure indicate that both induction and inhibition of glycometabolism could be beneficial for NAFLD? So what the exact process of glycometabolism is? Another example on FGF4, FGF4 inhibits apoptosis then what is next? Again, these things are described to reflect difficulty from a reader side. Hope the authors improve to make your manuscript suit for publication. 

- The author claimed that the manuscript was revised by a native English editor. However, I don't think the overall writing is better than the previous version. To illustrate using the first paragraph in the introduction "Once NAFLD is diagnosed, overall mortality is increased compared to nonNAFLD patients [9]. Cardiovascular disease (CVD) is the leading cause of death in people with NAFLD [10]. Furthermore, cancer-related mortality is among the leading causes of mortality in NAFLD patients [11]." So which one is the true leading cause: CVD or cancer? This is just an example. 

- A lot of ambiguous statement remains available. To illustrate using FGF21, first paragraph "It could protect people from obesity, insulin sensitivity, glucose/lipid metabolism, and vascular homeostasis in a certain extent [64], mainly induced by long-term fasting, ketogenic and high-carbohydrate diets, and free fatty acids (FFAs)." It is ok to protect obesity, but how it protects insulin sensitivity, metabolism, and vascular homeostasis? These should be "insulin resistance, aberrant metabolisms, and irregular vascular homeostasis"? Again, I could not point to every point, and hope that the authors would concern about them.

 - Figure 2 is just a repeat of Table 2.

Author Response

Honestly, I don't think that the manuscript is thoroughly improved. Here are some examples.

Response:

Thank you for your professional checking and critical comments, which is not only helpful to this manuscript. The manuscript has been revised following your suggestions. The changes were marked using built-in Track Changes in Microsoft Word and the main modifications are as following.

- Figure 1: I agree that it's so hard to describe everything in just a figure. That is the reason I suggest separating the figure. Also, it is fine to suggest the reader to find and search other publication to seek the detail. However, it is not fair to provide ambiguous or incorrect information. An example on glycometabolism, figure 1 indicate that FGF4 induces this process, but FGF19 inhibits it; but both of them inhibit NAFLD. Dose this figure indicate that both induction and inhibition of glycometabolism could be beneficial for NAFLD? So what the exact process of glycometabolism is? Another example on FGF4, FGF4 inhibits apoptosis then what is next? Again, these things are described to reflect difficulty from a reader side. Hope the authors improve to make your manuscript suit for publication.

Response:

Thanks to your comments, Figure 1 has been checked and improved accordingly.

For glycometabolism, FGF4 induces this process, and FGF19 inhibit CREB/PGC1A to alleviate the inhibition of CREB/PGC1A on glycometabolism. Thus, FGF4 and FGF19 are both beneficial to glycometabolism, to inhibit NAFLD.

In the lower part of Figure1, FGF4 is wrong and has been corrected to FGF1. FGF1 could inhibit apoptosis, to alleviate the process of NAFLD.

- The author claimed that the manuscript was revised by a native English editor. However, I don't think the overall writing is better than the previous version. To illustrate using the first paragraph in the introduction "Once NAFLD is diagnosed, overall mortality is increased compared to nonNAFLD patients [9]. Cardiovascular disease (CVD) is the leading cause of death in people with NAFLD [10]. Furthermore, cancer-related mortality is among the leading causes of mortality in NAFLD patients [11]." So which one is the true leading cause: CVD or cancer? This is just an example.

Response:

Thank you for your checking. These sentences have been revised to “The overall mortality of NAFLD patients is increased compared to non-NAFLD patients [9]. Cardiovascular disease (CVD) and cancer are the main causes of death in people with NAFLD [10,11].” Following your comments, the manuscript has been further checked and revised. The changes were marked in the revised manuscript.

- A lot of ambiguous statement remains available. To illustrate using FGF21, first paragraph "It could protect people from obesity, insulin sensitivity, glucose/lipid metabolism, and vascular homeostasis in a certain extent [64], mainly induced by long-term fasting, ketogenic and high-carbohydrate diets, and free fatty acids (FFAs)." It is ok to protect obesity, but how it protects insulin sensitivity, metabolism, and vascular homeostasis? These should be "insulin resistance, aberrant metabolisms, and irregular vascular homeostasis"? Again, I could not point to every point, and hope that the authors would concern about them.

Response:

Thank you for your detailed suggestion. It has been improved to “insulin resistance, aberrant metabolisms, and irregular vascular homeostasis” accordingly. The manuscript has been checked thoroughly and improved following your suggestion.

- Figure 2 is just a repeat of Table 2.

Response:

According to your comments, Figure 2 has been removed in the revised version of the manuscript.